# Oxygen Uptake On-Kinetics during Low-Intensity Resistance Exercise: Effect of Exercise Mode and Load

**DOI:** 10.3390/ijerph16142524

**Published:** 2019-07-15

**Authors:** Victor M. Reis, Eduardo B. Neves, Nuno Garrido, Ana Sousa, André L. Carneiro, Carlo Baldari, Tiago Barbosa

**Affiliations:** 1Research Centre in Sports Sciences, Health Sciences and Human Development, CIDESD, 5000-103 Vila Real, Portugal; 2Brazilian Army Research Institute of Physical Fitness, Rio de Janeiro 22291-090, Brazil; 3Department of Sciences of Physical Education & Sports, University Institute of Maia, ISMAI, 4475-690 Maia, Portugal; 4Department of Physical Education & Sports, State University at Montes Claros, Minas Gerais 39401-089, Brazil; 5eCampus University, 22060 Novedrate, Italy; 6Department of Physical Education & Sports Science, Nanyang Technologic University, Singapore 637616, Singapore

**Keywords:** VO_2_ kinetics, resistance exercise, energy cost

## Abstract

Oxygen uptake (VO_2_) kinetics has been analyzed through mathematical modeling of constant work-rate exercise, however, the exponential nature of the VO_2_ response in resistance exercise is currently unknown. The present work assessed the VO_2_ on-kinetics during two different sub maximal intensities in the inclined bench press and in the seated leg extension exercise. Twelve males (age: 27.2 ± 4.3 years, height: 177 ± 5 cm, body mass: 79.0 ± 10.6 kg and estimated body fat: 11.4 ± 4.1%) involved in recreational resistance exercise randomly performed 4-min transitions from rest to 12% and 24% of 1 repetition maximum each, of inclined bench press (45°) and leg extension exercises. During all testing, expired gases were collected breath-by-breath with a portable gas analyzer (K4b^2^, Cosmed, Italy) and VO_2_ on-kinetics were identified using a multi-exponential mathematical model. Leg extension exercise exhibited a higher R-square, compared with inclined bench press, but no differences were found in-between exercises for the VO_2_ kinetics parameters. VO_2_ on-kinetics seems to be more sensitive to muscle related parameters (upper vs. lower body exercise) and less to small load variations in the resistance exercise. The absence of a true slow component indicates that is possible to calculate low-intensity resistance exercise energy cost based solely on VO_2_ measurements.

## 1. Introduction

The oxygen uptake (VO_2_) on-kinetics has been shown to respond according to different workloads and types of exercise [1,2] as well as to aerobic fitness [3]. In 1982, Whipp and co-workers [4] presented a VO_2_ on-kinetics three-phase model, in which: (i) the cardio-dynamic phase (lasting ≈ 15–20 s) is due to the fast increase in alveolar O_2,_ (ii) the second phase (primary component lasting ≈ 2–3 min) is characterized by an exponential increase of VO_2_ due to muscle oxygen demand, and (iii) the third phase is described either by an additional slow rise on VO_2_ (slow component, SC), superimposing the primary component initiated at exercise, or by the stabilization in VO_2_ at the moderate exercise intensity domain [5]. 

The mathematical analysis of VO_2_ on-kinetics with multi exponential models enables assessment of the intensity at which the individual adjusts to the energy demand (through time constant parameter) as well as the possible existence of a SC. The VO_2_ on-kinetics has been described to be sensitive to the type of muscle contraction regime [6], pattern of motor unit recruitment [3], oxygen provision to the tissues [7], total muscle mass involved in exercise [8] and to body position [9]. To test these latter hypotheses, VO_2_ on-kinetics multi exponential modeling has been extensively described in running [6], cycling [10], and also in swimming exercise [11,12], and by simultaneously comparing different modes of exercise [13]. However, to the best of our knowledge, it has not yet been studied during resistance exercise (RE). 

For years resistance exercise was solely based on high-intensity, i.e., 70 % of 1 repetition maximum (1-RM) or above, but progressively it has been included in exercise programs designed to address body mass control and fat mass loss, which typically use intensities below 50% 1-RM. In fact, a recent study suggests that even at low intensities (12% and 24% of 1-RM), RE could comprise an energy cost between ∼5 kcal.min^−1^ and ∼12 kcal.min^−1^ [14]. However, the bioenergetics of RE are still poorly understood and both oxygen [15] and lactate-based methods of analysis [16] often lead to controversy [17]. Consequently, research on the energy expenditure involved in the execution of RE has increased exponentially, showing a large variation of results (from 2 to 11 kcal·min^−1^ in men and from 2 to 5 kcal·min^−1^ in women, which largely depend on the type of exercise and muscle groups that are involved [18]. However, it is presently unknown whether VO_2_ on-kinetics is sensitive or not to these muscle related parameters in RE, as shown previously for constant exercise. Hence, to investigate VO_2_ on-kinetics is warranted to aid the future design of standard protocols designed to assess energy cost during RE.

Due to the growing interest in low-intensity RE, as an efficient method to weight loss purposes, and/or to address the elderly and some pathologies, the aim of the present study was to assess the VO_2_ on-kinetics during two different sub-maximal RE—inclined bench press (45°) and leg extension. Considering that VO_2_ modeling was developed based on walking and running at constant velocity that predominantly utilized the lower body, it was hypothesized that leg extension exercise VO_2_ kinetics would be different compared with inclined bench press exercise VO_2_ kinetics. In addition, we test the hypothesis that small load variations (12% and 24% of 1-RM) would not affect the VO_2_ kinetics in-between exercises. The results of this study will have important implications in the exercise prescription for individuals involved in body mass loss or fat mass loss RE programs.

## 2. Materials and Methods

### 2.1. Participants

A total of 12 males (age: 27.2 ± 4.3 years, height: 177 ± 5 cm, body mass: 79.0 ± 10.6 kg and estimated body fat: 11.4 ± 4.1 %) engaged in RE training for at least one year with three or more training sessions per week, volunteered and were selected for this study. Individuals who used medication which could influence their cardiorespiratory response were not included in the sample. After medical approval, the volunteers received the explanations about the procedures, as well as the risks and discomforts involved in the study and signed the written consent form. All procedures were approved by the Institutional Review Board and were conducted according to the principles expressed in the Declaration of Helsinki. During the whole experimental period the participants were asked not to engage in any type of physical exercise.

### 2.2. Experimental Design

Data was collected over six testing sessions. In the first session, subjects’ height, weight, and skinfolds were collected by an experienced and ISAK certified technician and one-repetition maximum measurements recorded for the inclined bench press and leg extension (performed with trademark standardized machines—Panatta Sport, Apiro, Italy). Upper-body and lower-body exercises were chosen to target different body parts. In the second visit, seventy-two hours later, the 1-RM in inclined bench press and leg extension tests were repeated. From the third to the sixth visit (with 24 h intervals each), subjects performed four 4-min constant-intensity bouts for the inclined bench press and leg extension exercise at 12% and 24% of 1-RM (2 loads × 2 exercises). Exercise order for each individual was random, as was the intensity, and no warm-up was performed in any test. A recovery of 15 min was allowed between bouts. The cadence of 15 repetitions per minute (2 s on the eccentric and 2 s on the concentric phase) was paced by an electronic metronome sound (MA—30, Korg, Melville, NY, USA). All testing sessions were performed in the afternoon (except for the anthropometric measurements), at a temperature between 20–25 °C and 35–45% relative air humidity.

### 2.3. Measurements

Height, weight and seven skin folds (chest, mid-axillary, tricipital, sub scapular, abdominal, supra iliac, and thigh) were measured. A calibrated caliper (Lange, Cambridge Scientific Industries, Watertown, MA, USA) and a digital medical scale with stadiometer (Seca 763, Seca, São Paulo, SP, Brazil) were used for all measurements. In the 1-RM test, the highest 1-RM with less than 5% difference was considered as the true 1-RM. 

Pulmonary gas exchange parameters, particularly VO_2_, were calculated from the gas concentration and flow using a telemetric portable system (K4b^2^, Cosmed, Rome, Italy). Before each test, a reference air calibration of the device was performed using a gas sample with a 16% O_2_ and 5% CO_2_ concentrations, according to the manufactured instructions, as described elsewhere [19]. The data were telemetric displayed and analyzed breath-by-breath. The participants were asked not to perform any Valsala Maneuvers during exercise and were instructed to pace their breath frequency accordingly to the movement cadence.

### 2.4. VO_2_ Modeling

Prior to the VO_2_ kinetic modeling, a linear interpolation was applied to all breath-by-breath collected data to fit the time response to 1 s intervals. Thereon, an average filter with 11 samples (data ± 5 values) was used to smooth the data. According to the short-term fluctuations observed in the data, the number of samples used before and after the central value was chosen to eliminate the high-frequency oscillations and simultaneously to preserve the information from the original data. 

The kinetics of VO_2_ was modeled by the following exponential function:(1)$$\overset{\cdot }{V}O_{2}\left(t\right)=A_{0}+A_{1}\times \left(1-e^{-\left(\frac{t}{{tau}_{1}}\right)}\right)+A_{2}\times \left(1-e^{-\left(\frac{t-TD}{{tau}_{2}}\right)}\right)$$
where $\overset{\cdot }{V}O_{2}\left(t\right)$ represents the oxygen consumption per unit of time (ml∙min^−1^); A_0_ is the baseline value for the VO_2_ (mL∙min^−1^), A_1_ and A_2_ (mL∙min^−1^) are the amplitudes of the primary and slow component phases (respectively), TD is the time delay from the first exponential phase until the beginning of the second exponential (s), and tau_1_ and tau_2_ are time constants (s). A non-linear regression was applied to fit the time responses of VO_2_, in which coefficients were obtained by the least squares method. All mathematical procedures and modeling were done using MATLAB R2010b (Mathworks, Natick, MA, USA) for Windows^®^.

### 2.5. Statistical Analysis

Data distribution normality and sphericity were confirmed with the Shapiro-Wilk and Mauchly tests, respectively. To compare the parameters of the model according to the different intensities and different exercises, a two-way repeated-measures factorial ANOVA (exercise × intensity) was applied, with a post-hoc Bonferroni test performed. Magnitudes of standardized effects (η^2^) were determined against the following criteria: small, 0.2–0.5; moderate, 0.5–0.8, and large, >0.8. The significance level was set at 5% (*p* ≤ 0.05), with the results presented as means and standard deviations. All data analysis was conducted using SPSS (17.0, Science, IBM, Chicago, IL, USA) for Windows^®^.

## 3. Results

Figure 1 shows the VO_2_ on-kinetics of one representative subject, in the four experimental conditions. 

Table 1 presents the VO_2_ on-kinetics parameters, extracted from a bi-exponential model during the four experimental sessions. The repeated measures analysis of variance revealed a significant effect of the exercise factor in the goodness curve fit R-square (*p* = 0.04, η^2^ = 0.38) and in Adjusted R-square (*p* = 0.039, η^2^ = 0.39) in the leg extension exercise when compared with inclined bench press exercise (Table 2). No differences were detected among exercises as to the sum square error (*p* = 0.279, η^2^ = 0.128) and the root mean square error (*p* = 0.981, η^2^ = 0). No significant effect of intensity and interaction (exercise × intensity) were detected among others variables of the bi-exponential classical model.

## 4. Discussion

The aim of the present study was to assess, for the first time, the VO_2_ on-kinetics during two different sub-maximal RE—inclined bench press (45°) and leg extension, at two different intensities (12 vs. 24% of 1-RM). It was reported that a significant effect of the exercise was found in the robustness of the model, as leg extension exercise exhibited a higher R-square and adjusted R-squared with no differences between exercises found in any other model parameter. This did not confirm the first hypothesis that leg extension exercise VO_2_ kinetics would be different compared with inclined bench press VO_2_ kinetics. Moreover, neither load variations (12% and 24% of 1-RM) nor interaction of load and exercise type affected the adjustment of the models in both exercises, confirming the secondary hypothesis. 

To the best of our knowledge, the VO_2_ on-kinetics on RE was not previously investigated through mathematical modeling in the low-intensity domain. A single study [20] described 10 s average VO_2_ during constant-load at various RE, but the breath-by-breath mathematical modeling was not used and analysis was limited to off-kinetics (i.e., post-exercise). An important methodological point when using a multi exponential model is the starting value of the baseline VO_2_. No significant differences were found between any interventions in the baseline VO_2_ in the present study, suggesting that the significant effect of the exercise in the robustness of the model was, indeed, due to variables related with the exercise itself. In fact, a variable that could influence the robustness of the curve fit model (both R-square and in adjusted R-square) is the total amount of muscle mass involved in the exercise. 

The body position adopted during exercise has also been shown to influence VO_2_ kinetics in constant exercise [12,21,22]. In fact, the supine position has been shown to impair muscle perfusion pressure, which generally results in a slower VO_2_ kinetics [23]. Moreover, the amount of work that is performed by the upper vs. lower body seems to influence not only the time constant, but as well as the SC during constant exercise. In fact, it has been shown that heart rate and VO_2_ kinetics were slower, and the SC higher, in arm cranking compared with cycling exercise performed within the high-intensity domain [24]. The present study demonstrated that RE for upper body or lower body exercise with a load variation (12% and 24% of 1-RM) did not modify VO_2_ kinetics. A possible explanation to the lack of differences found could be the low intensity in which the RE was performed. In fact, some exploratory data reported lactate threshold to be around 30% 1-RM in leg press, bench press and biceps curl exercises [25]. Therefore, it could be possible that the low intensity performed did not involve a sufficient stimulus to promote differences in the VO_2_ kinetics, irrespective of the differences in body segments—upper vs. lower body muscles [5]. This latter trend does not corroborate previous studies conducted at intensities above the lactate threshold, where differences in the VO_2_ kinetics are expected to occur. Collectively, these results seem to suggest that VO_2_ on-kinetics is not clearly sensitive to muscle related parameters in RE at intensities under the lactate threshold, as shown previously for continuous exercise at higher intensities, and therefore, we cannot clearly confirm the hypothesis as to exercise type influence on VO_2_ on-kinetics in RE exercise at these specific intensities. 

The VO_2_ on-kinetics has also been shown to vary according to the exercise intensity [1,2], with the appearance of the SC typically reported for exercise above the individual´s lactate threshold [26]. Although changes in VO_2_ on-kinetics according to the exercise intensity or the appearance of the SC are not yet established for RE, one might expect SC to also be more prominent at intensities above the 4 mM^−1^ lactic threshold. It has been suggested that during RE this threshold occurs somewhere close to 30% of 1RM intensity [25]. Hence, we tested the subjects at intensities below this latter threshold, aiming to confirm if the bi-exponential model fits the data without a prominent SC. The SC, in this study, was assessed through mathematical modeling which is more precise and accurate than calculating the increase in VO_2_ between the second or third minute and the last minute of exercise [27,28]. The fact that we did not see the development of a true SC confirms that exercise intensities herein were steady-state, and sustains the threshold previously proposed [25]. Previous studies with high-intensity knee-extension exercise also reported values for VO_2_ SC amplitude at 190.00 ± 80.00 mL∙min^−1^ [29] and at 120.00 ± 7.00 mL∙min^−1^ [7]. These values are higher than those in the present study, although they still fall below the typical threshold for a true SC [30]. In short, our results suggest that during low-intensity RE the SC is absent, as previously described for other modes of exercise [31]. This absence of SC indicates that for low-intensity resistance exercise it is possible to estimate its energy cost based on VO_2_ measurements. The energy cost herein, was ~5 kcal·min^−1^ and ~6.5 kcal·min^−1^ in the inclined bench press, respectively at 12% and at 24% of the 1-RM value. In the leg extension it was ~6 kcal·min^−1^ and ~9 kcal·min^−1^, respectively. These values are between typical reports for walking at 4 km.h^−1^ (~4 kcal·min^−1^) or jogging at 8 km.h^−1^ (~12 kcal·min^−1^) for a subject of similar body mass [32].

## 5. Conclusions

The mathematical model used in the present study showed that the leg extension exercise fit better to model when compared with inclined bench press. The VO_2_ on-kinetics seems to be more sensitive to muscle related parameters (upper vs. lower body exercise) and less to small load variations (12% vs. 24%) in RE. Moreover, it reported an absence of a SC at low-intensity inclined bench press and leg extension exercises at intensities up to 24% 1-RM. We suggest that the bioenergetics of low-intensity RE (probably below the lactic threshold for this type of exercise) may match typical observations for whole-body constant exercise (i.e., running, cycling or swimming), though with likely less goodness of fit. The absence of a true SC indicates that is possible to calculate low-intensity resistance exercise energy cost based solely on VO_2_ measurements. Hence, there is potential to enable a calorie count during low-intensity resistance exercise through portable and wearable devices, as currently often used in typical aerobic exercise.

## Figures and Tables

**Figure 1 ijerph-16-02524-f001:**
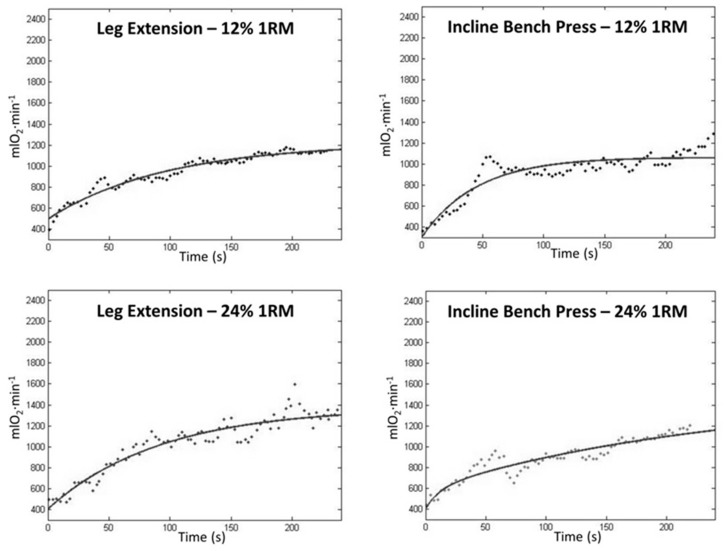
Oxygen uptake on-kinetics in the four conditions of exercise. The fit line is shown.

**Table 1 ijerph-16-02524-t001:** Mean (standard deviation) of VO_2_ on-kinetics parameters extracted from the bi-exponential model in each exercise and intensity.

Model Parameter	Leg Extension		Inclined Bench Press	
	12% 1-RM	24% 1-RM	12% 1-RM	24% 1-RM
A_0_, mL∙min^−1^	511.47 (87.75)	567.08 (126.37)	459.38 (131.11)	483.01 (124.40)
A_1_, mL∙min^−1^	988.38 (326.31)	942.20 (227.73)	787.43 (305.80)	874.45 (217.18)
A_2_, mL∙min^−1^	32.18 (135.24)	14.53 (141.72)	48.01 (123.91)	−16.91 (109.64)
tau_1_, s	96.22 (40.10)	123.25 (45.39)	123.23 (87.06)	116.90 (102.63)
tau_2_, s	7.86 (10.24)	14.00 (13.70)	6.24 (5.85)	29.41 (50.48)
TD, s	74.53 (111.82)	87.54 (165.94)	86.00 (130.11)	111.63 (225.52)
R^2^	0.80 (0.13)	0.76 (0.10)	0.68 (0.18)	0.65 (0.21)
Adj. R^2^	0.79 (0.14)	0.74 (0.12)	0.66 (0.19)	0.62 (0.22)
SSE	1.41 × 10^6^ (1.99 × 10^6^)	1.56 × 10^6^ (1.77 × 10^6^)	1.01 × 10^6^ (9.62 × 10^5^)	1.25 × 10^6^ (1.90 × 10^6^)
RMSE	107.83 (48.80)	127.65 (62.16)	110.85 (44.43)	128.33 (77.11)

Note: A_0_ = baseline oxygen uptake; A_1_ = amplitude of the 1st exponential (fast component); A_2_ = amplitude of the 2nd exponential (slow component, given by the mathematical modeling); tau_1_ and tau_2_ = time constants of the equation for the 1st and 2nd exponentials, respectively; TD = time delay of the second exponential; Adj = adjusted; SSE = sum square error; RMSE = root mean square error.

**Table 2 ijerph-16-02524-t002:** Results from repeated measures analysis of variance to the oxygen kinetics parameters extracted from the bi-exponential model, considering exercise type and intensity as factors.

Parameters	Exercise			Intensity			Exercise × Intensity		
	F	*p* Value	η^2^	F	*p* Value	η^2^	F	*p* Value	η^2^
A_0_	5.073	0.051	0.360	0.184	0.678	0.020	0.092	0.768	0.010
A_1_	1.574	0.241	0.149	0.182	0.680	0.020	0.265	0.619	0.029
A_2_	0.293	0.602	0.031	2.735	0.133	0.233	0.726	0.416	0.075
tau_1_	0.137	0.720	0.015	0.179	0.682	0.019	0.494	0.500	0.052
tau_2_	1.180	0.306	0.116	3.016	0.116	0.251	1.746	0.219	0.162
TD	0.136	0.721	0.015	0.247	0.631	0.027	0.062	0.809	0.007
Adj. R^2^	5.487	0.044 *	0.392	0.246	0.632	0.034	0.405	0.540	0.045
R^2^	5.796	0.039 **	0.379	0.316	0.588	0.027	0.423	0.531	0.043
SSE	1.325	0.279	0.128	0.009	0.928	0.001	2.475	0.150	0.216
RMSE	0.001	0.981	0.000	0.826	0.387	0.084	0.718	0.419	0.074

Note: A_0_ = baseline oxygen uptake; A_1_ = amplitude of the 1st exponential (fast component); A_2_ = amplitude of the 2nd exponential (slow component, given by the mathematical modeling); tau_1_ and tau_2_ = time constants of the equation for the 1st and 2nd exponentials, respectively; TD = time delay of the second exponential; Adj = adjusted; SSE = sum square error; RMSE = root mean square error. **p* < 0.05; ***p* < 0.01.

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
