# Peer review of "Oxygen Uptake On-Kinetics during Low-Intensity Resistance Exercise: Effect of Exercise Mode and Load"

_ijerph, 2019, doi:10.3390/ijerph16142524_

Round 1
Reviewer 1 Report
The manuscript, as presented, is in need of extensive English editing. More importantly, the rationale of the study needs to be further explained (hopefully not developed). As a resistance training practitioner, researcher, etc. I am having a hard time pulling the rationale of the work out of the manuscript. Ok, no one has done this... ok, this is something that was measured cleanly and expertly... now why? If I had to give this my best academic effort, I would say that this study is establishing a model for the calculation of resistance training energy expenditure using VO2 on-kinetics. This is not how the introduction, results, discussion read, however. It is almost like the study is written as if the 12% and 24% 1RM comparison is the goal... other than the establishement of a model for measurement, the experimental conditions used have no relevance to exercise, sport, etc. The overall message of the manuscript needs to be reoworked.
Introduction- no specific comments on introduction, save the critique above regarding a clearer explanation of the rationale, use of references and logical build of the introduction is here... rework the emphasis
Materials and methods- explanations of the measurement, modeling, and statistical analysis are excellent
Line 84- explain to the audience why a multi-joint upper body exercise and a single-joint lower body exercise were chosen as representative movements, would these have different amounts of muscle mass?; 1RM measurement should not be undertaken in single-joint movements; further, if postural differences in exercises are considered a means of altering VO2 on-kinetics then why would you choose and upper- and lower-body exercise that had different postures?
Line 87- When is a 4-minute set of resistance training at 12% or 24% of 1RM utilized? Again, if the intent of the study is the development of a working model for measurement, then this can be explained.
Line 96- consistent investigator for skinfolds? reliability?
Line 105- controlling breathing rate during VO2 measurement might drastically change the measurement, no?
Results- so the leg extension exercise data points fit the 'goodness curve' better than the inclined bench press exercise... what would this mean for the use of a particular exercise as a model of energy expenditure? why is this important? is this important? what is the ultimate goal here?
Discussion-
Line 179- what evidence do you have that the quadriceps (Leg Extension) comprise a greater muscle mass than the (multi-joint) pectoralis major, triceps, anterior deltoid, etc muscles?
Author Response
REVIEWER 1
Comments and Suggestions for Authors
The manuscript, as presented, is in need of extensive English editing. More importantly, the rationale of the study needs to be further explained (hopefully not developed). As a resistance training practitioner, researcher, etc. I am having a hard time pulling the rationale of the work out of the manuscript. Ok, no one has done this... ok, this is something that was measured cleanly and expertly... now why? If I had to give this my best academic effort, I would say that this study is establishing a model for the calculation of resistance training energy expenditure using VO2 on-kinetics. This is not how the introduction, results, discussion read, however. It is almost like the study is written as if the 12% and 24% 1RM comparison is the goal... other than the establishement of a model for measurement, the experimental conditions used have no relevance to exercise, sport, etc. The overall message of the manuscript needs to be reoworked.
We have re-written some parts to improve the style of writing.
The study does not aim itself to establish a model for the calculation of energy expenditure during resistance exercise (RE). However, the study provides important support for submaximal exercise protocols that anyone may want to apply when estimating energy expenditure. In fact, any ramp protocol that may be used at RE requires previous knowledge about the O2 on-kinetics. This has been extensively studied and described for cyclical movement (i.e.running or cycling), but not for RE. Hence, this study contributes to future development of such protocols but it does not propose a protocol itself. As stated in the conclusions of the manuscript, we have shown that leg exercise is preferable compared with upper-body exercise for this purpose. We have also shown that 4-min below the lactic threshold for RE may be used in ramp RE protocols due to the lack of evident slow-component (SC). On a more practical setting, the lack of SC also points out that these low-intensities may be used in RE volume training when weight loss, rather than neuromuscular adaptations, are warranted. Indeed, some modern trends with RE such as, circuit training, functional training, body-pump, etc,, do involve low-intensity RE performed during several minutes with low loads.
Due to space limitation the rationale above cannot be fully inserted into the paper. However, we have placed a new phrase in the introduction, attempting to clarify this issue.
Introduction- no specific comments on introduction, save the critique above regarding a clearer explanation of the rationale, use of references and logical build of the introduction is here... rework the emphasis
Materials and methods- explanations of the measurement, modeling, and statistical analysis are excellent
Line 84- explain to the audience why a multi-joint upper body exercise and a single-joint lower body exercise were chosen as representative movements, would these have different amounts of muscle mass?; 1RM measurement should not be undertaken in single-joint movements; further, if postural differences in exercises are considered a means of altering VO2 on-kinetics then why would you choose and upper- and lower-body exercise that had different postures?
We have done pilots with more exercises (a total of 10 exercises). The exercises herein were chosen because they showed good stability as to the O2 on-kinetics. And they were also chosen to contrast different body segments. The trunk position, which is key to O2 kinetics is not that different as trunk inclination in the inclined bench press was 45 and it was around 60 degrees in the leg extension exercise. No exercise was performed with the trunk fully vertical nor horizontal.
Due to space limitation this explanation cannot be fully inserted into the paper. However, we have placed a new phrase in the methods, attempting to clarify this issue.
Line 87- When is a 4-minute set of resistance training at 12% or 24% of 1RM utilized? Again, if the intent of the study is the development of a working model for measurement, then this can be explained.
We believe that above we have explained this rationale.
Line 96- consistent investigator for skinfolds? Reliability
Height, weight, and skinfolds were collected by an experienced and ISAK certified technician. This was now included.
Line 105- controlling breathing rate during VO2 measurement might drastically change the measurement, no?
Valsalva maneuver or forced breaths do disturb the O” stability and when they are performed O2 does not measures true aerobic energy cost. Hence, the control of breathing is warranted,
Results- so the leg extension exercise data points fit the 'goodness curve' better than the inclined bench press exercise... what would this mean for the use of a particular exercise as a model of energy expenditure? why is this important? is this important? what is the ultimate goal here?
As explained above and referred in the conclusions, these results favor the use of leg-exercise rather than upper-body exercise when low-intensity steady-state RE is warranted to assess energy cost of exercise.
Discussion-
Line 179- what evidence do you have that the quadriceps (Leg Extension) comprise a greater muscle mass than the (multi-joint) pectoralis major, triceps, anterior deltoid, etc muscles?
You are right. We cannot conclude on the muscle mass differences. We have deleted from the manuscript the phrase where a larger muscle mass was claimed to be involved in the leg-extension exercise (second paragraph of discussion).
Reviewer 2 Report
The authors should be applauded for a well designed and conducted study. The authors attempted to quantify the VO2 kinetics during two different modes of resistance training with two different intensities. Overall, this is a less studied subject and this work adds to the literature. I will detail out minor changes below line by line.
Line 15: please re-write this first paragraph for clarity. Suggested is " Oxygen uptake (VO2) kinetics has been analysed through mathematical modelling of constant work-rate exercise, however, currently unknown is the exponential nature of the VO2 response in resistance exercise"
Line 19: please change m to cm
Line 24: Change L to Leg
Line 41: "exponential models enables" Please re-write this it enables who or what?
Line 43: change sensible to sensitive
Line 50-51: Re-write this sentence it is unclear on the true meaning.
Line 55-58: Should the numbers have a decimal between them "2.7-11 and 2.3-5.2? Change "elicited" to involved
Line 63-66: change "were" to was "walk" to walking and change "constant exercise" to at a constant velocity or speed
Line 72: convert m to cm
Line 82: Data was instead of "Data were"
Line 84: measurements recorded for the incline bench press in stead of "in incline"
Line 118: need a space before "A non-linear"
Line 132: change on kinetics to on-kinetics
Line 140: Remove the word "Though" just leave No differences
Line 142: change to (exercise x intensity)
Table 2: change the p value to italics
Table 2: you should consider adding magnitude of effects for all variables not just the ones you listed below the table. I would remove those two values and add them to the whole table.
Table 1 and 2: below the table you should write "Note:" followed by all the definitions for clarity
Line 159: re-write this sentence for clarity you used intensity twice in the same sentence
Line 173: Your A0 baseline oxygen uptake for exercise was close to a significant effect. In order for this statement to stand you need to provide an effect size showing a trivial or small effect. \
Line 187: re-write "heavy exercise domain" for clarity do you mean the load was heavy or the intensity was high? currently unclear.
line 185: change "expectable" to expected
Line 218: please re-write this sentence
Author Response
REVIEWER 2
Comments and Suggestions for Authors
The authors should be applauded for a well designed and conducted study. The authors attempted to quantify the VO2 kinetics during two different modes of resistance training with two different intensities. Overall, this is a less studied subject and this work adds to the literature. I will detail out minor changes below line by line.
Line 15: please re-write this first paragraph for clarity. Suggested is " Oxygen uptake (VO2) kinetics has been analysed through mathematical modelling of constant work-rate exercise, however, currently unknown is the exponential nature of the VO2 response in resistance exercise"
It was changed.
Line 19: please change m to cm
It was changed.
Line 24: Change L to Leg
It was changed.
Line 41: "exponential models enables" Please re-write this it enables who or what?
It was changed.
The mathematical analysis of VO2 on-kinetics with multi exponential models enables assessment of the intensity at which….
Line 43: change sensible to sensitive
It was changed.
Line 50-51: Re-write this sentence it is unclear on the true meaning.
It was changed.
For years resistance exercise was solely based on high-intensity (i.e. 70 % 1-RM or above) but progressively it has been included in exercise programs designed to address body mass control and fat mass loss, which typically use intensities below the 50% 1-RM
Line 55-58: Should the numbers have a decimal between them "2.7-11 and 2.3-5.2? Change "elicited" to involved
Both changes performed.
Line 63-66: change "were" to was "walk" to walking and change "constant exercise" to at a constant velocity or speed
It was changed.
Line 72: convert m to cm
It was changed.
Line 82: Data was instead of "Data were"
It was changed.
Line 84: measurements recorded for the incline bench press in stead of "in incline"
It was changed.
Line 118: need a space before "A non-linear"
It was changed.
Line 132: change on kinetics to on-kinetics
It was changed.
Line 140: Remove the word "Though" just leave No differences
It was changed.
Line 142: change to (exercise x intensity)
It was changed.
Table 2: change the p value to italics
It was changed.
Table 2: you should consider adding magnitude of effects for all variables not just the ones you listed below the table. I would remove those two values and add them to the whole table.
It was changed.
Table 1 and 2: below the table you should write "Note:" followed by all the definitions for clarity
It was changed.
Line 159: re-write this sentence for clarity you used intensity twice in the same sentence
It was changed.
Line 173: Your A0 baseline oxygen uptake for exercise was close to a significant effect. or small effect.
We understand, but since no significant effect was detected must we really change the text?
Line 187: re-write "heavy exercise domain" for clarity do you mean the load was heavy or the intensity was high? currently unclear.
Changed to “High-intensity domain”
line 185: change "expectable" to expected
It was changed.
Line 218: please re-write this sentence
It was changed to:
In short, our results suggest that during low-intensity RE the SC is absent, as previously described for other modes of exercise [31]. This absence of SC indicates that for low-intensity resistance exercise it is possible to estimate its energy cost based on VO2 measurements.
Reviewer 3 Report
Major concern
- From Fig 1 and Table 1, there apparently seem to be some differences between data from 12% and 24% in both exercises. For example, the graphs in the 24% looked much stepper than that in the 12%. Also, the values were generally much higher in the 24% relative to 12%. But the authors reported that there were no statistical significances in many of the variables measured between these 2. Is it possible that the lack of significant difference could due to lack of sample size? Could authors have a check on their coefficient of variation of the oxygen measurements and then re-analyse their data on whether because of the large variations observed that there were no mathematical significant differences between the 2 different intensities and/or exercises.
Minor concerns:
- There were several spelling mistakes throughout the manuscript. There are Line 24, Line 181, Line 185, please check through thoroughly .
- Line 56-57. The values ins not clear, it is 2.7 or 2 and 7. Please use (.) rather than (,) for decimal point. And also use (zero) to ensure that the values are clear. For example, use 2.0 rather than 2.
- Line 59. Need a reference for this statement.
- Line 195. ‘expected’ rather than ‘expectable’.
- Line 219. ‘fit’ rather than “adjusted”.
- Authors could provide some information on the caloric cost of the 2 exercises at the 2 different intensity – and perhaps provide a simple comparison between these 2 exercises with some common exercises used to burn body fats such continuous low to moderate intensity walking or jogging – to provide some more practical applications of their findings.
Author Response
REVIEWER 3
Comments and Suggestions for Authors
Major concern
- From Fig 1 and Table 1, there apparently seem to be some differences between data from 12% and 24% in both exercises. For example, the graphs in the 24% looked much stepper than that in the 12%. Also, the values were generally much higher in the 24% relative to 12%. But the authors reported that there were no statistical significances in many of the variables measured between these 2. Is it possible that the lack of significant difference could due to lack of sample size? Could authors have a check on their coefficient of variation of the oxygen measurements and then re-analyse their data on whether because of the large variations observed that there were no mathematical significant differences between the 2 different intensities and/or exercises.
We agree that the small sample size can account for the lack of statistical significance.
We are sorry but we could not understand the alternative analysis that the reviewer has suggested.
Minor concerns:
- There were several spelling mistakes throughout the manuscript. There are Line 24, Line 181, Line 185, please check through thoroughly .
It was changed.
- Line 56-57. The values ins not clear, it is 2.7 or 2 and 7. Please use (.) rather than (,) for decimal point. And also use (zero) to ensure that the values are clear. For example, use 2.0 rather than 2.
It was changed.
- Line 59. Need a reference for this statement.
This refers to reference [18], which is now properly identified.
- Line 195. ‘expected’ rather than ‘expectable’.
It was changed.
- Line 219. ‘fit’ rather than “adjusted”.
It was changed.
- Authors could provide some information on the caloric cost of the 2 exercises at the 2 different intensity – and perhaps provide a simple comparison between these 2 exercises with some common exercises used to burn body fats such continuous low to moderate intensity walking or jogging – to provide some more practical applications of their findings.
In the final part of the discussion we added a paragraph to address this request.
Round 2
Reviewer 1 Report
I appreciate your focused, calculated responses to my queries and believe that my major considerations have been addressed. My concerns about the rationale have been fixed, and I can now more clearly see the relevance of the study. I am definitely from the 70% or higher school of resistance exercise, and perhaps this is why I was a bit hesitant with the rationale at first. Thank you for expanding my perspective on this. Best wishes with future studies.